organic chemistry/medicinal chemistry/ synthetic chemistry

natural products, synthesis, acaricidal activity, *Varroa destructor*

**Author for correspondence:**
H. E. Katerinopoulos
e-mail: kater@chemistry.uoc.gr

This article has been edited by the Royal Society of Chemistry, including the commissioning, peer review process and editorial aspects up to the point of acceptance.

# Enantioselective synthesis of a costic acid analogue with acaricidal activity against the bee parasite *Varroa destructor*

S. Georgiladaki, D. Isaakidis, A. Spyros, G. K. Tsikalas and H. E. Katerinopoulos

Department of Chemistry, University of Crete, Voutes Campus, 71003 Heraklion, Crete, Greece

(iD) HEK, 0000-0003-2187-0825

One major disease of the pupae and the adult bee is the so-called *Varroosis* that is owing to the bee parasite *Varroa destructor*. It is an ectoparasite of bees, causing significant losses in the bee population needed for honey production as well as for pollination in agriculture. Costic acid is a sesquiterpene-carboxylic acid present in the plant *Dittrichia viscosa*. Recent studies by our group have shown that costic acid acts as acaricide against *V. destructor*. Oxalic acid is also an acaricide commonly used against varroa mites. In spite of its structural simplicity—it is the simplest bicarboxlic acid—it is equipotent to costic acid which consists of a *trans*-decalin system with three chiral centres. The basic goal of this project was to design and synthesize a hybrid entity, incorporating aspects of both oxalic acid and costic acid that would be more active than the parent compounds. This approach introduces a useful strategy for the preparation of congeners of bioactive compounds and proposes a structural framework for a new series of acaricidal agents.

## 1. Introduction

A major disease of the pupae and the adult European bee is *Varroosis* that is owing to the bee parasite *Varroa destructor* [1,2]. Varroa is an ectoparasite of bees causing significant losses in the bee population that is needed not only for honey production but also for pollinating [3] an estimated $8–$10 billion worth of crops (for a recent commentary on the issue of pollinators, see [4]). Oxalic acid is a natural product that to date is considered one of the most efficient acaricidal agents used against

**Scheme 1.** The basic idea on the preparation of the synthetic congener.

*V. destructor* [5–8]. The compound is non-toxic to bees; however, its application suffers from a number of setbacks including possible detrimental effect on brood development [9–11] and the possible presence of oxalotrophic bacteria on varroa-attacked bees [12]. Costic acid (β-costic acid) is a natural sesquiterpene (for synthetic methodology towards costic acid, see [13,14]) present in many plant species including *Dittrichia viscosa*, a ruderal, perennial shrub common throughout the Mediterranean Basin [15]. The compound has been isolated by our group and assayed for its acaricidal activity [16,17]. It has been reported recently that α-costic acid, the double bond isomer of costic acid, also acts as acaricide against varroa [18].

Oxalic acid is the simplest bicarboxlic acid in nature. Costic acid bears a number of stereochemical restrictions imposed by its *trans*-decalin system and its three stereogenic centres. Yet, both compounds exhibit very similar acaricidal activity against *V. destructor*. This equipotency prompted us to test the acaricidal profile of an analogue incorporating structural elements of both moieties in its carbon framework (scheme 1).

# 2. Material and methods

## 2.1. General procedures

General procedures that were followed have been published elsewhere [19]. All reactions were carried out under anhydrous conditions in dry solvents, using argon or nitrogen in flame-dried glassware. Reactions were monitored by thin-layer chromatography (TLC) using silica gel plates from Merck (60F254), which were visualized under an ultraviolet-visible Lamp (254 nm and 366 nm, respectively) or with a 7% ethanolic solution of phosphomolybdic acid. Flash column chromatography was performed in silica gel 60 from Merck (230–400 mesh). Nuclear magnetic resonance (NMR) spectra were taken on an AMX500 Bruker Fourier transform-NMR (FT-NMR) or an MSL300 Bruker FT-NMR spectrometer; proton chemical shifts are reported in relative to tetramethylsilane. Mass spectra were acquired using sonic-spray ionization (SSI) mass spectrometry (MS). Absolute mass spectra were taken on a LTQ ORBITRAP XL with ETD-Thermo Fisher Scientific (San Jose, CA, USA; Bremen, Germany) with ion source: electrospray ionization (ESI) positive mode and an orbitrap mass analyser. The enantiomeric excess of the compounds was determined by high-performance liquid chromatography (HPLC) using DAICEL columns with chiral static phase (Chiralpak AS-H) used as mobile phase hexane and a detection wavelength of 240 nm. Gas chromatography (GC) analysis on a chiral column was also performed using a GC-2014 SHIMADZU instrument with a JW Chiral CP-Chrasil Dex CB column, a helium flow of 1.40 ml min$^{-1}$ at a temperature of 150°C. Optical rotation values were measured on the P3000 (A. Krüss Optronic) polarimeter. Specific rotation values, $[\alpha]_D^T$, refer to a concentration of 100 g ml$^{-1}$ and the polarimeter cell length was 50 mm. The symbol D refers to the D-line of sodium (589 nm) and the temperature ($T$) is given in degrees celsius (°C).

## 2.2. Synthetic procedures

The synthesis of compounds **1–7** is not described in this section as the preparation of **1** is known ([20] and the literature cited therein), and the synthetic procedures and/or spectral identification of compounds **2–7** is described in the following procedures.

### 2.2.1. Preparation of compounds *trans*-**2** and *trans*-**2a**

To a flame-dried three-necked 500 ml flask, 50 ml of dry diethyl ether and 200 mg of compound (4aR/S)-**1** (1.22 mmol) were added. Ammonia gas was added to the system and liquefied at −78°C (2-propanol /dry ice) under vigorous stirring, for 15 min. The excess of ammonia was restrained by a silicon oil

trap. Then 55 mg Li was added portionwise under argon gas. Gradually, the solution acquired a blue colour, characteristic of the reaction. After 15 min stirring, 368 mg $NH_4Cl$ was gradually added. When the solution was completely decolourized, 50 ml $H_2O$ was added and the mixture was extracted with ether and 5% aqueous HCl solution. The resulting mixture contained the desired ketone *trans*-2 and the corresponding alcohol *trans*-2a. Two hundred milligrams of this mixture were dissolved in 1.5 ml of acetone. Then, 253 µl of 8 N Jones reagent was added, at ice bath temperature. The solution was stirred at 0°C for 3 h, till the Jones oxidation was complete. The excess of Jones reagent was quenched by the addition of isopropyl alcohol and the mixture was extracted with ether and $H_2O$. The product was isolated from the crude mixture by flash chromatography on silica gel using ethyl acetate (EA): petroleum ether (PE) = 1 : 9 as eluent. The reaction yield was 176 mg (87%). Diastereomeric excess was found by GC analysis to be 92.8%. Chiral GC analysis of the major diastereomer indicated the presence of a racemic mixture.

### 2.2.2. Preparation of compounds *trans*-4a,b

To a 25 ml round bottom flask, ketone *trans*-2 (100 mg, 0.60 mmol) and dimethyl succinate (130 mg, 0.90 mmol) were transferred. Benzene (3 ml) was added and the mixture was distilled in a rotary evaporator, and then vacuum pumped for 1 h to remove traces of moisture.

In a flame-dried 25 ml round bottom flask equipped with condenser, 5 ml of dry *tert*-butanol was added and the system temperature was increased to 30°C. Then Na (30 mg) was added in small portions, and the system was heated to 90°C until the metal dissolved. The two solutions were transferred to an autoclave and were stirred at 100–110°C for 20 h. The solution was then neutralized by the addition of 6 N HCl (2 ml) and extracted with dichloromethane (3 × 20 ml). The organic layer was washed with saturated NaCl solution (3 × 10 ml), dried with anhydrous $MgSO_4$ and the solvent was removed *in vacuo*. The products were isolated from the crude mixture by flash chromatography on silica gel using EA : PE = 1 : 4 as eluent. The reaction yield was 134 mg (70%).

### 2.2.3. Preparation of compounds *trans*-5a,b

To a flame-dried two-necked 25 ml flask equipped with condenser, *trans*-4a,b (90 mg, 0.28 mmol) dissolved in 1 ml dry methanol was added. Then the system was cooled to 0°C and concentrated $H_2SO_4$ (0.009 ml, 0.13 mmol) was added. The solution was stirred at 0°C for 30 min and the temperature was gradually increased to room temperature. Then, the solution was heated at 70°C for 2 h. The solution was extracted with diethyl ether (3 × 20 ml) and the organic layer was washed with saturated NaCl solution (3 × 10 ml), dried with anhydrous $MgSO_4$, and the solvent was removed *in vacuo*. The product was isolated from the crude mixture by flash chromatography using 25% EA in PE as eluent. The reaction yield was 61 mg (65%).

### 2.2.4. Preparation of compound *trans*-6

To a flame-dried two-necked 25 ml flask, *trans*-5a,b (90 mg, 0.26 mmol) dissolved in 9 ml dry ethanol and $PtO_2$ (30 mg) was added. The solution was stirred for 1 h under the pressure of gaseous $H_2$. The catalyst was deactivated by the addition of dichloromethane and was removed from the solution by filtration on celite. The solvent was removed *in vacuo* to give pure *trans*-6. The reaction yield was 86 mg (98%). GC analysis of *trans*-6 on a non-chiral column indicated the presence of the two major diastereomers with a de of 67.8%. HPLC analysis on chiral column indicated the presence of two enantiomeric pairs, both as racemic mixtures.

### 2.2.5. Preparation of compound *trans*-7

To a flame-dried two-necked 25 ml flask, *trans*-6 (50 mg, 0.17 mmol) dissolved in 2 ml of dry dimethyl sulfoxide (DMSO) and *t*-BuOK (80 mg, 0.68 mmol) was added. The reaction was completed in 2 h. The addition of diethyl ether precipitated the product in its potassium salt form. The solid was isolated by centrifugation, dissolved in dry acetonitrile, transferred to a flame-dried flask and protonated using Amberlyst 15. The solvent was removed *in vacuo*. The reaction yield was 45 mg (98%).

## 2.2.6. Preparation of compound 10

A Dean-Stark water separator and condenser were applied to a 250 ml round bottom flask containing a magnetic stirrer. 2-Methylcyclohexanone ((±)8) (20.00 ml, 164.9 mmol) and (S)-(−)-α-methylbenzylamine (9) (21.2 ml, 164.9 mmol) were dissolved in 100 ml of dry toluene. The flask was placed in an oil bath and heated to 125°C under reflux conditions and argon gas for 24 h and kept overnight in agitation. A GC-mass spectrometry (GC-MS) spectrum revealed that the desired intermediate was formed at 93%. The solvent was removed *in vacuo*. The crude product was used directly in the next reaction.

## 2.2.7. Preparation of compound 11

To a flame-dried round-bottomed 250 ml flask, compound 10 (28.40 g, 131.92 mmol) dissolved in 75 ml of tetrahydrofuran was added. The flask was placed in an ice bath and 3-butene-2-one (mvk) was added (11.55 ml, 138.50 mmol) dropwise and under vigorous stirring. After the addition was complete, the ice bath was maintained for 30 min and then the solution remained stirred for 5 days at 20°C. This reaction required special conditions because of the convenience of mvk producing polymerization products. Finally, the solution was extracted with diethyl ether (3 × 20 ml) and the organic layer was washed with saturated NaCl solution (3 × 10 ml) and dried with anhydrous $MgSO_4$, and the solvent was removed *in vacuo*. The crude product was used directly in the next reaction.

## 2.2.8. Preparation of compound 12

To a 250 ml round-bottomed flask equipped with a magnetic stirring bar and placed in an ice bath, compound 11 was added (38.00 g) with 4 ml of acetic acid, which had previously been diluted in 16 ml of deionized $H_2O$ (20% solution). The solution was kept in an ice bath for 15 min and then, vigorously stirred for 2 h. The solution was extracted with $CH_2Cl_2$ (3 × 20 ml) and the organic layer was washed with saturated NaCl solution (3 × 10 ml) and dried with anhydrous $MgSO_4$, and the solvent was removed *in vacuo*. It is worth mentioning that in the GC-MS spectrum, the bicyclic unsaturated ketone 4aR-1, which was expected to form in a subsequent reaction via Robinson's cyclization, was already in the largest proportion as it is possible that the hydrolysed amine acted as a base to form the compound.

## 2.2.9. Preparation of compound 4aR-1

To a flame-dried 25 ml round bottom flask, 5 ml of dry methanol and 320 mg of Na cut into small pieces were transferred. The addition of Na was at 0°C while gradually the solution formed came to room temperature. To a second pre-dried 250 ml round bottom flask, compound 12 (12 g, 65.90 mmol) dissolved in 70 ml of dry methanol was transferred. The temperature of this solution was lowered to 0°C and the sodium methoxide solution, which was also at 0°C, was gradually added. The solution was allowed to gradually come to 25°C and then heated to 70°C for 20 h. The solution was neutralized by the addition of acetic acid to pH7 and was extracted with $CH_2Cl_2$ (3 × 20 ml) and the organic layer was washed with saturated NaCl solution (3 × 10 ml) and dried with anhydrous $MgSO_4$, and the solvent was removed *in vacuo*. The product was isolated from the crude mixture by flash chromatography on silica gel using ether : PE = 1 : 4 as eluent. The desired 4aR-1 product was formed in 37% (4.09 g). The optical rotation of the substance was determined as $[\alpha]^{20}_D = -238°$ (EtOH, $c = 1.0$) corresponding to 99% ee according to the literature data [20].

## 2.2.10. Preparation of compound (4aR,8aR)-2

To a 500 ml three-necked round-bottomed flask, 600 mg, (3.65 mmol) of compound (4aR)-1 in 50 ml of diethyl ether was transferred. Maintaining vigorous stirring, ammonia gas was added to the solution, and was liquefied and mixed with the ether solution by means of a dry ice-2-propanol bath which kept the temperature of the system at −78°C. Ammonia gas was introduced into the solution for about 15 min, while its excess was collected by a silicone oil trap. The liquid ammonia flow was then discontinued; an inert gas supply was connected to the flask, and the addition of 170 mg of Li was added in portions. Gradually, the solution became blue, characteristic of the specific reaction (solvated electron). After 15 min, 368 mg of $NH_4Cl$ was added portionwise. When the solution was completely decolourized, 50 ml of $H_2O$ was added, and the system was extracted with ether and 5% aqueous HCl. The resulting mixture contained the desired ketone (4aR,8aR)-2 as well as the corresponding

alcohol. The whole mixture (200 mg) was subjected to Jones oxidation: it was dissolved in 1.5 ml of acetone, cooled in an ice bath, and then 1.0 ml of 8 N Jones reagent was added to the acetone solution. The system temperature was maintained at 0°C for 3 h, whereupon the reaction was complete. Excess Jones reagent was neutralized by the addition of isopropyl alcohol and the mixture was extracted with ether and $H_2O$. Separation of the mixture was accomplished by column chromatography, on silica gel, using as eluent EA : PE = 1 : 9. The yield of the reaction was 79% (480 mg). The optical rotation of the new product was determined as $[\alpha]^{20}_D = -42.0°$ (EtOH, $c = 1.0$). GC analysis of (**4a***R***,8a***R***)-2** using chiral column indicated an ee of 90.8% (see the electronic supplementary material).

## 2.2.11. Preparation of compound (4a*R*,8a*R*)-4a,b

In a 25 ml round bottom flask, (**4a***R***,8a***R***)-2** (280 mg, 1.70 mmol) and dimethyl succinate (380 mg, 2.60 mmol) were transferred. An amount of 2–3 ml of benzene was added thereto, and the mixture was distilled off on a rotary evaporator and then left in the vacuum pump for about 1 h to remove any solvent or moisture. To another flame-dried 25 ml round bottom flask equipped with a condenser, 10.0 ml of dry *tert*-butanol was transferred and the temperature was raised to 30°C (to avoid solidification of *tert*-butanol (mp 25–26°C). To this flask, sodium (840 mg) cut into small pieces was then added and the system was heated to 90°C until the metal was dissolved. Then, the above two solutions were transferred to an autoclave and heated by means of an oil bath to 100–110°C for 20 h. The solution was then neutralized by the addition of 2 ml of 6 N HCl and extracted with dichloromethane (3 × 20 ml) and the organic layer was washed with saturated NaCl solution (3 × 10 ml). The organic layer was dried with anhydrous $MgSO_4$ and the solvent was removed using a rotary evaporator. The mixture was separated by column chromatography on silica gel eluting with EA : PE = 1 : 4. It should be noted that the main reaction product was the methyl ester. The desired products were formed in 70% (357 mg).

## 2.2.12. Preparation of compound (4a*R*,8a*R*)-5a,b

To a flame-dried, 25 ml two neck round bottom flask equipped with a condenser was transferred 90 mg (0.28 mmol) of the reactant dissolved in 1 ml of dry methanol. The system was then cooled to 0°C and then 9.0 µl concentrated $H_2SO_4$ was added in one portion. The solution remained for 30 min under stirring and gradually returned to ambient temperature. The solution was then heated under stirring to 70°C for 2 h. The solution was extracted with diethyl ether (3 × 20 ml) and the organic layer was washed with saturated NaCl solution (3 × 10 ml). The organic layer was dried with anhydrous $MgSO_4$ and the solvent was removed in a rotary evaporator. The mixture was separated by column chromatography on silica gel eluting with EA : PE = 1 : 4. It is worth noting that the only product was the dimethyl ester (**4a***R***,8a***R***)-5a,b** (65%, 54 mg).

## 2.2.13. Preparation of compound (2*R*,4a*R*,8a*R*)-6

In a flame-dried 25 ml two-necked round-bottomed flask, 90 mg (0.31 mmol) of the mixture (**4a***R***,8a***R***)-5a,b** dissolved in 9.0 ml of dry ethanol and 30 mg of the catalyst ($PtO_2$) were transferred. The solution remained for 1 h, stirring under $H_2$ atmosphere. The catalyst was inactivated by the addition of dichloromethane and removed from the solution by filtration through celite. The solvent was removed using a rotary evaporator. The yield of the reaction was 98% (89 mg). GC analysis of (**2***R***,4a***R***,8a***R***)-6** on a non-chiral column indicated the presence of two diastereomers with a de of 91.9%. HPLC analysis on chiral column indicated the presence of one enantiomer identified by its retention time as one of the two enantiomers of the major diastereomer in ***trans*-6.**

## 2.2.14. Preparation of compound (2*R*,4a*R*,8a*R*)-7

To a pre-dried, 25 ml two neck round bottom flask, (**2***R***,4a***R***,8a***R***)-6** (50 mg, 0.17 mmol) dissolved in 2.0 ml of dry DMSO and 80 mg (0.68 mmol) of potassium *tert*-butoxide were added to the system. In 2 h, the reaction mixture contained the final product as well as the corresponding monoacids, whereupon it was allowed to stir for 10 additional hours to complete the reaction. Then, the addition of diethyl ether precipitated the product in the di-potassium salt form. Centrifugation of this solution removed the DMSO from the salt, which was then dissolved in dry acetonitrile, transferred to a

pre-dried flask and protonated using Amberlyst 15. The solvent was removed *in vacuo*. The yield of the reaction was 98% (45 mg).

## 2.3. Mites and bees

The mites and bees studies protocol described below is a modification of the procedure published earlier [16,17]. The mites (*V. destructor*) were collected from colonies of *Apis mellifera* with sister queens. For mite collection, two different methods were used: in method A, an apparatus introduced by Ariana *et al.* [21] was used. Approximately 1000 infected adult honeybees were transferred directly from bee frames into a wire-screen cylinder. Then the cylinder was placed inside a second Plexiglas cylinder and $CO_2$ was released for 5 min with a flow rate of $5 \, l \, min^{-1}$ causing anaesthesia to the mites (as well as to the bees). The whole apparatus was then shaken several times to separate the mites from the bees. The inner cylinder was taken out of the apparatus to return the bees to their mother colony. A few minutes later, they all recovered from the effect of the anaesthesia. In this way, more than 80% of mites were separated from the bees and fell to the bottom of the outer cylinder. The bottom lid of the outer cylinder was taken off and the mites were collected and placed into the test vials. The second method involved removal of the mites from infected adult bees using a soft brush with the help of a stereoscope. In either case, no mortality of bees was observed after repeated trials and the mortality of mites observed was less than 0.1%.

## 2.4. Screening tests

Screening of the synthetic analogues was performed as described previously in the study of costic acid [16,17]. The experiment was conducted in a completely randomized design under laboratory conditions in five replications. With the help of a stereoscope, the isolated varroa was immediately placed in groups of five at the bottom of 35 ml glass vials. The compounds studied were used to make acetone solutions, with a concentration of $10 \, mg \, ml^{-1}$, which were placed on filter paper fitted to the caps. Measurements were made using 60 µl doses, chosen as the optimal dose for costic acid activity [16,17]. In some vials, 60 µl of acetone was applied as control (five replicates). Acetone was removed with nitrogen gas from the vials, to which 20 µl $H_2O$ was added to maintain the necessary moisture levels. The vials were then sealed with varroa and left in an incubator at 25°C. Mortality of mites was recorded under a stereoscope binocular set at 2 h time intervals.

## 2.5. Data analysis

Graphs and statistical analysis using one-way ANOVA followed by the Neuman–Keuls test were done using GRAPHPAD PRISM (v. 5.03) [22].

# 3. Results and discussion

In order to satisfy the needs of a structure–activity relationship study on the product, three independent syntheses were followed. A non-stereoselective synthesis involved a modified Heathcock procedure of the Robinson annulation of 2-methyl cyclohexanone with methyl vinyl ketone that yielded the racemic unsaturated decalone **1** (scheme 2). Catalytic hydrogenation of **1** in the presence of 10% Pd/C yielded a mixture of *cis/trans* 4a-methyl-2-decalone (**2**) in a 1 : 3 ratio, as indicated by the relative integrations in the $^1$H-NMR spectra of the two diastereomers.

Given that the structure of oxalic acid prevents any carbon substitution in the molecule, we chose succinic acid as a proper dicarboxylic substitute. Stobbe condensation of **2** with dimethyl succinate in BuOH/BuONa gave the expected monomethyl ester **3** as the minor product, together with 70% of the *t*-butyl ester **4** as a mixture of olefinic isomers. The free carboxylic moiety in **4** was esterified in MeOH/$H_2SO_4$ to give diester **5** which was hydrogenated to yield **6**. Deprotection of both ester moieties was achieved by treatment of **6** with t-BuOK/DMSO that gave the desired product **7** as a mixture of diastereomers in 98% yield. The same procedure was followed for **3** giving the respective products in similar yields.

A second, diastereoselective, synthesis of **7** was achieved via hydrogenation of the double bond of racemic **1** in Li/NH$_3$(liq) yielding *trans* 4a-methyl-2-decalone (**trans-2**) together with the corresponding *trans*-decalol **8** that was readily converted to the decalone via Jones oxidation giving

**Scheme 2.** Non-stereoselective synthesis of costic acid analogue **7**.

**Scheme 3.** Diastereoselective synthesis of costic acid analogue **trans-7**.

**trans-2** in a total 87% yield (scheme 3). Again, Stobbe condensation of **trans-2** with dimethyl succinate in BuOH/BuONa furnished the *t*-butyl ester **trans-4** as a mixture of olefinic isomers in 70% yield. Similarly, conversion of **trans-4** to diester **trans-5** (65% yield), catalytic (10% Pd/C) hydrogenation of the double bond to give **trans-6** (98%) and ester deprotection with t-BuOK/DMSO gave the desired product **trans-7** as a mixture of diastereomers in 98% yield.

The enantioselective synthesis of **7** required the enantioselective construction of the decalin system securing the (R) configuration of the 4a-stereogenic centre in the 2-decalone system. The enantioselective Robinson annelation took place in the presence of S-(−)-proline (abbreviated as S-(−)**9**) yielding (**4aR**)-**1** in 37% overall yield via intermediate **11** (scheme 4). The optical rotation of the compound was $[\alpha]^{20}_D = -238°$ (EtOH, $c = 1.0$) a value that corresponds to 99% ee, based on the data by Revial & Pfau [20]. A reduction in Li/NH₃(liq) gave the *trans*-decalone system in 80% overall yield. In this case, given the presence of the (**4aR**) stereocentre, the diastereoselectivity of the reaction was translated in enantioselectivity securing the desired stereochemistry in the decalone system (**4aR,8aR**)-**2**. The optical rotation of the new product was determined as $[\alpha]^{20}_D = -42.0°$ (EtOH, $c = 1,0$). Stobbe reaction gave (**4aR,8aR**)-**4** as a mixture of the two isomeric olefins (70% yield) and subsequent esterification furnished the diester (**4aR,8aR**)-**5** in 65% yield.

The next challenge in the synthesis was the reduction of the double bonds in the isomeric olefin system (scheme 5). Our approach to the problem was the following. Because the side chain in costic acid assumes an equatorial conformation, it would be very likely that a simple catalytic

**Scheme 4.** Enantioselective approach to costic acid analogue **7**; synthesis of intermediate decalone **(4a*R*)-1v.**

**Scheme 5.** Enantioselective approach to costic acid analogue **7**; synthesis of **(2*R*,4a*R*,8a*R*)-7**.

hydrogenation of **(4a*R*,8a*R*)-5** would lead to a similar conformation yielding the most stable costic acid analogue. To our satisfaction, catalytic hydrogenation of the olefinic mixture yielded the saturated compound in 98% yield. Should our assumption be correct, the product would be **(4a*R*,8a*R*,2*R*)-6** with the axial H-2 proton at the same side of the decalin system as the axial H-8a proton. Indeed, the NOESY spectrum showed a clear interaction between the two protons (see the electronic supplementary information) confirming the fact that the absolute stereochemistry in the three stereogenic centres in 6 was the same as in the parent compound. The optical rotation of purified **(4a*R*,8a*R*,2*R*)-6** was ($[\alpha]^{20}_D = -4.1$ ($c = 1.8$, EtOH). The protective groups were removed with t-BuOK/ DMSO, yielding the desired product **(4a*R*,8a*R*,2*R*)-7** ($[\alpha]^{20}_D = +23$ ($c = 0.5$, EtOH)), in 98% yield.

At this point, it should be noted that the Stobbe reaction created a fourth chiral centre on the side chain, which is α to the carboxyl moiety. Unfortunately, the synthetic strategy that was followed and the nature of the final products do not allow the determination of the absolute configuration of the α-stereocentre; Stobe reaction conditions are not enantioselective. Moreover, the α proton in **(2*R*,4a*R*,8a*R*)-6** and **(2*R*,4a*R*,8a*R*)-7** is very acidic and this fact precludes any attempts to modify the structures in strong basic or acidic conditions. For instance, the use of Mosher ester analysis would require the reduction of the carboxylates to the corresponding alcohols under conditions that would epimerize this stereocentre. We believe that the active configuration of the α-centre would be revealed through an enantioselective synthesis of a *tricyclic analogue* that would lead to the formation of one enantiomer. However, we believe that **(2*R*,4a*R*,8a*R*)-7** incorporates in its structure enough common structural elements with costic acid to secure acaricidal activity.

The three diacids were assayed for acaricidal activity on varroa mites [16,17] using costic acid as a positive control. Costic acid was tested first and exhibited exactly the same activity pattern as reported

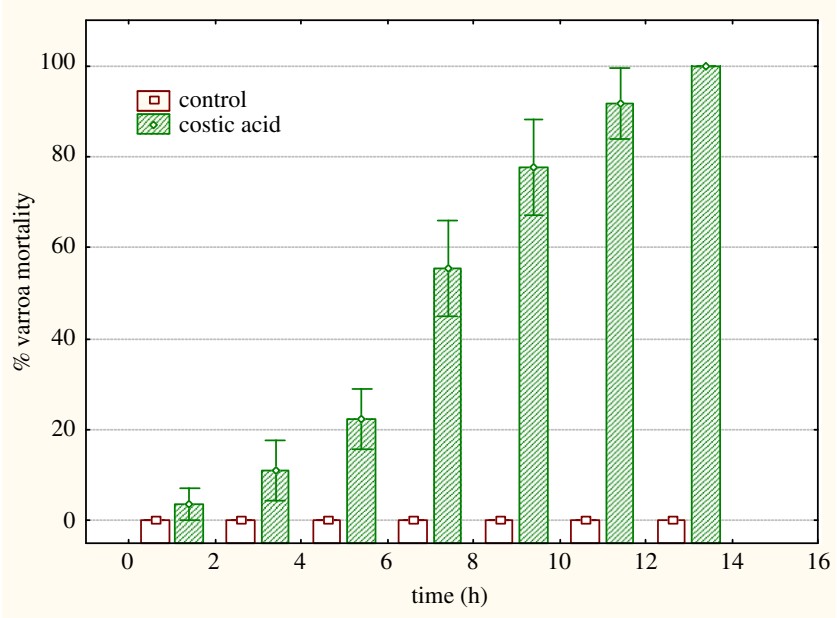

**Figure 1.** Per cent mortality of *V. destructor* upon treatment with costic acid.

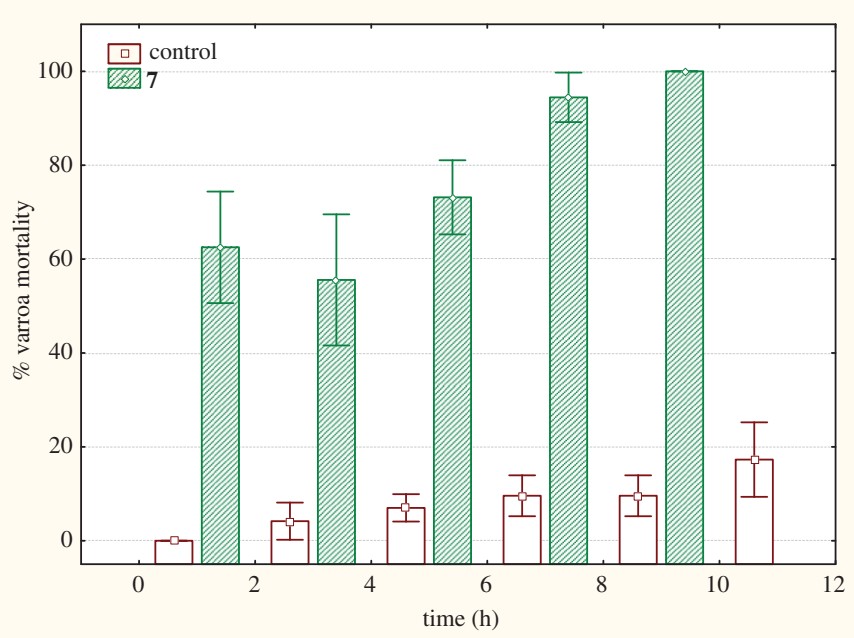

**Figure 2.** Per cent mortality of *V. destructor* upon treatment with compound **7**.

previously [16,17]. The compound shows a distinct activity after 6 h of incubation of the mites at beehive temperatures, reaching 100% mortality rates at 14 h of mite exposure to the drug (figure 1).

Compound **7**, which is a mixture of diastereomers appeared to be more potent than the parent compound maintaining 50–60% varroa mortality rates the first 2–4 h and reaching maximum mortality after 10 h (figure 2). Control values did not exceed 20%.

The diastereoselectively prepared costic acid analogue *trans*-7 was also active against the mite; in this assay, varroa mortality (approx. 60%) was achieved within 1 h and the maximal effect was observed within 5 h, indicating that the *trans* disposition of the ring fusion is probably a structural element necessary for the drug activity (figure 3).

The results of the last assay were quite impressive: 2 h application of **(4a*R*,8a*R*,2*R*)-7** to varroa under conditions imitating beehive environment resulted in 80% death in the mites. In 6 h, varroa mortality

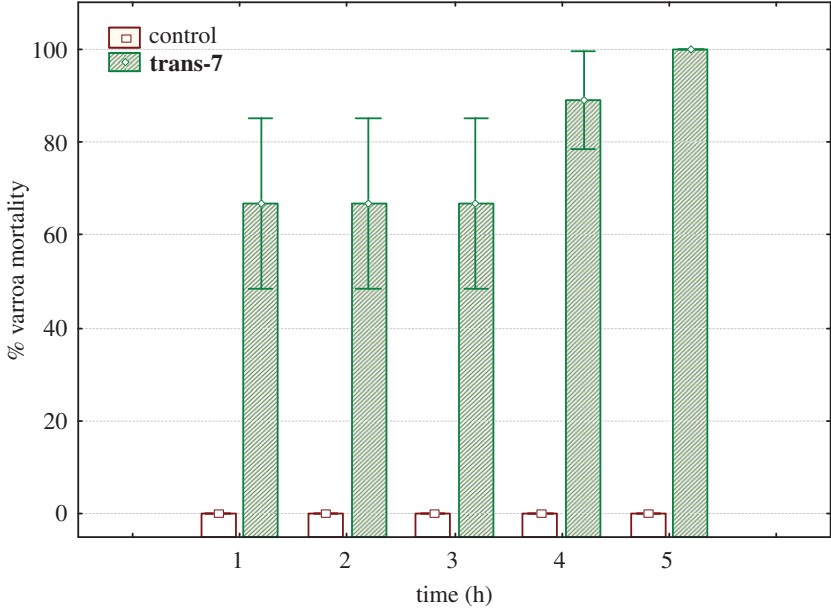

**Figure 3.** Per cent mortality of *V. destructor* upon treatment with compound **trans-7**.

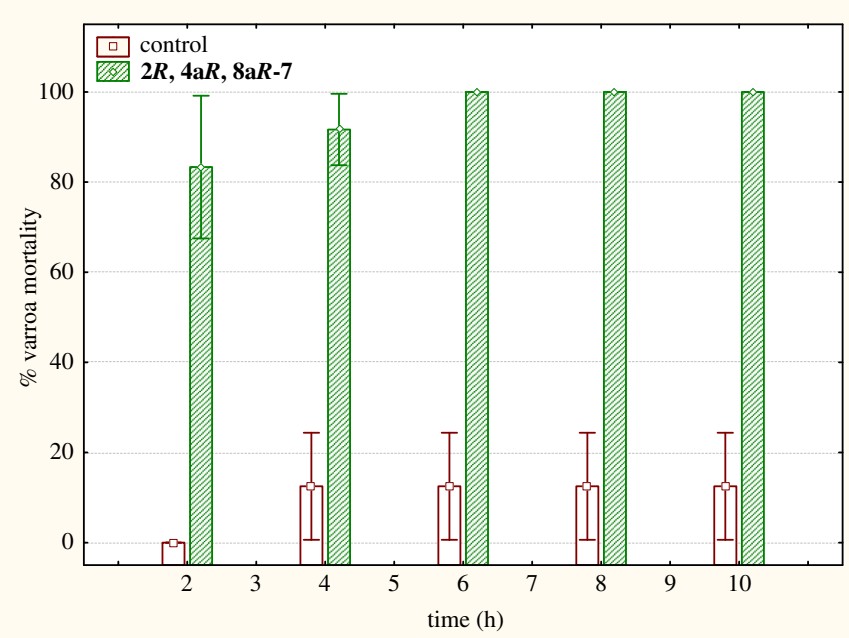

**Figure 4.** Per cent mortality of *V. destructor* upon treatment with compound **(4a*R*,8a*R*,2*R*)-7.**

reached 100%, whereas the corresponding value for the isomeric mixture **7** was 20% (figure 4). The parent compound, costic acid caused 10% and 55% varroa death in 2 and 8 h, respectively, and its activity reached 100% death rate after 14 h, in agreement with earlier findings. Control values did not exceed 15% during the 10 h period.

## 4. Conclusion

Costic acid is a sesquiterpene present in many plant species including *Dittrichia viscosa*. The compound has been isolated by our group, assayed for its acaricidal activity and found to be active against the bee parasite *V. destructor*. The fact that α-costic acid, which is the double bond isomer of costic acid also acts against varroa, supported the idea that the carbon framework of this bicyclic system might be essential

for activity. Oxalic acid is also an acaricide that is widely used against varroa. In spite of its considerable structural differences with costic acid, it is equipotent to the latter in its acaricidal activity. It was hypothesized that a hybrid entity, incorporating aspects of both oxalic acid and costic acid, would be more active than the parent compounds. The target compound was synthesized diastereoselectively and enantioselectively using classic, well-established chemistry, and acaricidal activity studies showed that the enantioselectively prepared product bearing the bicyclic system of costic acid and a bis-carboxylic side chain exhibited higher activity than the parent compounds. Although the idea was shown to be valid, the question on how each of the two parent compounds acts still remains. A possible answer could be drawn from comparative studies on the mechanism of action of costic acid, oxalic acid and **(4aR,8aR,2R)-7** on varroa mites, that are ongoing in our laboratory.

Data accessibility. This article does not contain any additional data.

Authors' contributions. S.G. performed the synthesis of the compounds; D.I. performed the studies on varroa; A.S. did the data analysis and drafted the diagrams; G.K.T. participated in the spectral and physical properties analysis; H.E.K. had a major contribution in conception of the study, data interpretation, supervision of the project and writing of the manuscript. All authors gave final approval for publication.

Competing interests. There are no conflicts of interest to declare.

Funding. This work has been supported by the ULTRACHIRAL project of the Horizon 2020 Framework Programme for Research and Technological Development, as well as project no. 2018EP40200000—SAEP 402 from the Regional Government of Crete.

Acknowledgements. We thank the ProFI (Proteomics Facility at IMBB-FORTH) for performing all the HRMS analyses. We also thank V. Giannopoulos, at the laboratory of Prof. I. Smonou and Dr M. Sofiadis, University of Crete, for de and ee studies.

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
