## [Reviewer comments · Royal Society Open Science]

Review History

RSOS-200612.R0 (Original submission)

Review form: Reviewer 1

Is the manuscript scientifically sound in its present form?

Yes

Are the interpretations and conclusions justified by the results?

Yes

Is the language acceptable?

Yes

Do you have any ethical concerns with this paper?

No

Have you any concerns about statistical analyses in this paper?

No

Recommendation?

Accept with minor revision (please list in comments)

Comments to the Author(s)

I have attached my comments as a PDF file (Appendix A).

Review form: Reviewer 2

Is the manuscript scientifically sound in its present form?

Yes

Are the interpretations and conclusions justified by the results?

Yes

Is the language acceptable?

Yes

Do you have any ethical concerns with this paper?

No

Have you any concerns about statistical analyses in this paper?

No

Recommendation?

Accept with minor revision (please list in comments)

Comments to the Author(s)

The Authors describe the synthesis and investigation of acaricidal activity of several hybrid compounds derived from costic and oxalic acid. Generally, the synthesis is reasonable and the biological results encouraging. In order to improve this manuscript, may I draw the Authors attention to the following points:

- The Authors describe this chemistry as an enantioselective synthesis, but there does not seem to be an ee for all chiral compounds - this needs to be rectified.
- Yields should be included in schemes
- HRMS data should be given over low resolution mass spec
- LCMS traces should be given as a further indication of purity (or elemental analysis)
- The structure of compound 12 does not seem to be correct
- No need to include the characterisation data for the compounds in the manuscript - these belong in the SI

Decision letter (RSOS-200612.R0)

Dear Dr Katerinopoulos:

Title: Enantioselective Synthesis of a Costic Acid Analog with Acaricidal Activity against the Bee Parasite *Varroa destructor*
Manuscript ID: RSOS-200612

Thank you for submitting the above manuscript to Royal Society Open Science. On behalf of the Editors and the Royal Society of Chemistry, I am pleased to inform you that your manuscript will be accepted for publication in Royal Society Open Science subject to minor revision in accordance with the referee suggestions. Please find the reviewers' comments at the end of this email.

The reviewers and handling editors have recommended publication, but also suggest some minor revisions to your manuscript. Therefore, I invite you to respond to the comments and revise your manuscript.

Because the schedule for publication is very tight, it is a condition of publication that you submit the revised version of your manuscript before 04-Jul-2020. Please note that the revision deadline will expire at 00.00am on this date. If you do not think you will be able to meet this date please let me know immediately.

Kind regards,
Dr Laura Smith

Publishing Editor, Journals

On behalf of the Subject Editor Professor Anthony Stace and the Associate Editor Dr Andrew Harned.

RSC Associate Editor:

Comments to the Author:

I agree with the referees that this is a nice piece of work that is very appropriate for this journal and our readership. They have both offered relevant comments/criticisms that should be addressed in a revised manuscript. In particular I would like to draw the author's attention to the concerns raised by Reviewer 2 regarding compound characterization. These are valid concerns that should be addressed. If an LCMS is not readily available, providing copies of ¹HNMR spectra would be a valid alternative to showing compound purity. Many of the compounds may not be readily detected by LCMS.

RSC Subject Editor:

Comments to the Author:

(There are no comments.)

Reviewer comments to Author:

Reviewer: 1

Comments to the Author(s)

I have attached my comments as a PDF file.

Reviewer: 2

Comments to the Author(s)

The Authors describe the synthesis and investigation of acaricidal activity of several hybrid compounds derived from costic and oxalic acid. Generally, the synthesis is reasonable and the biological results encouraging. In order to improve this manuscript, may I draw the Authors attention to the following points:

- The Authors describe this chemistry as an enantioselective synthesis, but there does not seem to be an ee for all chiral compounds - this needs to be rectified.
- Yields should be included in schemes
- HRMS data should be given over low resolution mass spec
- LCMS traces should be given as a further indication of purity (or elemental analysis)
- The structure of compound 12 does not seem to be correct
- No need to include the characterisation data for the compounds in the manuscript - these belong in the SI

Author's Response to Decision Letter for (RSOS-200612.R0)

See Appendix B.

Decision letter (RSOS-200612.R1)

Dear Dr Katerinopoulos:

Title: Enantioselective Synthesis of a Costic Acid Analog with Acaricidal Activity against the Bee Parasite *Varroa destructor*
Manuscript ID: RSOS-200612.R1

It is a pleasure to accept your manuscript in its current form for publication in Royal Society Open Science. The chemistry content of Royal Society Open Science is published in collaboration with the Royal Society of Chemistry.

On behalf of the Subject Editor Professor Anthony Stace and the Associate Editor Dr Andrew Harned.

RSC Associate Editor
Comments to the Author:
The authors have addressed the concerns raised by the previous review. I can recommend publication at this time.

Reviewer(s)' Comments to Author:

Appendix A

ARTICLE REVIEW

Title: “Enantioselective Synthesis of a Costic Acid Analog with Acaricidal Activity against the Bee Parasite *Varroa destructor*”

Journal: *Royal Society of Open Science* (published in collaboration with RSC)

Authors: Georgiladaki, Sofia; Isaakidis, Demosthenes; Spyros, Apostolos; Tsikalas, George; Katerinopoulos, Haralambos, University of Crete Department of Chemistry

Summary:

- In previous work, costic acid was isolated and assayed to determine its acaricidal activity; its similar biological activity to oxalic acid prompted the design a hybrid structure and exploration of its acaricidal profile (with the hypothesis that a molecule incorporating structural features of both acids would have increased activity).
- The synthesis relies on classic approach: all reactions are published in the literature and considered standard procedures for classic reactions (e.g. Stobbe condensation, catalytic hydrogenation, dissolving metal reduction, Robinson annulation).
- Biological evaluation was an extension of previously reported protocols (Beilstein J. Org. Chem. 2017, 13, 952–959. doi:10.3762/bjoc.13.96) involving the collection of infected honeybees and temporary carbon dioxide-induced anesthesia. The bees and mites are then separated for testing with no harm to the bees.
- Screening tests were performed based on previously reported protocols (Beilstein J. Org. Chem. 2017, 13, 952–959. doi:10.3762/bjoc.13.96); mites were incubated at 25 degrees Celsius under necessary moisture levels in the presence of compound, which was administered via filter paper dosed with 60 μ L of acetone solution.
- Three diacids were assayed using costic acid as a positive control: compound **7**, *trans*-**7**, and (4 α R,8 α R,2R)-**7**; all three proved to be as efficacious, achieving 100% death, as costic acid but resulted in greater percentage of death in shorter timeframes compared to the control.
- Overall, the authors’ hypothesis seems to be supported by these preliminary results.

Critical Evaluation:

- The work appears to be appropriate for the broad scope and quality of the journal.
- There are several important issues regarding clarity that need to be addressed.
 - It seem plausible that in addition to the alkene isomers shown (**3a**, **3b**, **4a**, **4b**), that the conjugated acrylate and fumarate isomers are plausible. These would be access by a similar isomerization mech *via* enolate intermediates. Qualitatively, it seems that the acrylate/fumarate would also be thermodynamically preferred.
 - The α -stereocenter is not defined in any of the structures (**4**, **5**, **6**, **7**). Further, in cases where a diastereomeric ratios can be determined, it should be. Further,

inspection of the NMR such as certain versions of **6** and **7** appear to be single diastereomers. Thus, the stereochemistry at the α -position needs to be defined.

- While not ultimately prohibitive to the reader's understanding of the text, the numerous grammatical hamper the reading progression... please closely edit this manuscript an additional time before resubmission.
- The work demonstrates how classic, well-established chemistry (e.g. Claisen condensations, dissolving metal reduction, Robinson annulation) can be employed to access novel, valuable frameworks for preparing valuable lead molecules.

Recommendation:

- Moderate revisions are necessary prior to acceptance

Comments and Edits:

Page #	Section	Text/Figure/Scheme	Correction/Comment
1	Summary	“Cotic acid is a sesquiterpenecarboxylic acid, present in the plant Dittrichia viscosa .”	Remove comma
1	Summary	“...it is equipotent to costic acid which consists of a transdecalin system with 3 three chiral centers.”	Change “that” to “which” and “3” to “three”
1	Summary	“The basic idea in goal of this project was to design and synthesize a hybrid entity, incorporating aspects of both oxalic acid and costic acid, that would be is that, in spite of the significant differences in structural complexity, the combination of a simple dicarboxylate moiety with the carbon framework of costic acid would lead to an active acaricide. The hypothesis turned out to be correct: the synthetic product was more active than the parent compounds. This approach introduces a useful idea in strategy for the preparation of congeners of bioactive compounds, and in the same time	“The basic goal of this project was to design and synthesize a hybrid entity, incorporating aspects of both oxalic acid and costic acid, that would be more active than the parent compounds. This approach introduces a useful strategy for the preparation of congeners of bioactive compounds and proposes a structural framework for a new series of acaricidal agents.”

		proposes the a structural framework of a new series of acaricidal agents.”	
1	Introduction	Sentences 1 and 2	Nearly identical to those of the summary
1	Introduction	“varroosis”	Varroosis
1	Introduction	“an estimated \$8 to \$10 billion worth of crops, only to mention the USDA estimates.”	“an estimated \$8 to \$10 billion worth of crops.” (the last phase does not make sense)
1	Introduction	“alpha-costic acid, the double bond isomer of costic acid is also acts acting as acaricide against varroa”	“alpha-costic acid, the double bond isomer of costic acid also acts as acaricide against varroa”
1	Introduction	“analog incorporating structural element of both moieties”	“analog incorporating structural elements of both moieties”
2	Synthetic Procedures		Faint grey highlighting throughout experimental section?
4	Synthetic Procedures	Preparation of compound 10:	2-methylcyclohexanone is defined in parentheses as “mvk”, this is incorrect
6	Mites and Bees	“For mite collection two different methods were used: In method A an apparatus introduced by Ariana et al [19] was used.”	“For mite collection two different methods were used: in method A an apparatus introduced by Ariana et al [19] was used.”
6	Mites and Bees	“The inner cylinder was taken out of the apparatus to return the bees to their mother colony; a few minutes after they had recovered from the effect of the anesthesia.”	“The inner cylinder was taken out of the apparatus to return the bees to their mother colony a few minutes after they had recovered from the effect of the anesthesia.”
6	Mites and Bees		It is not clear how the two methods of mite collection are different as the text states that soft brushes and stereoscopes are used to separate the mites and bees in both cases
6	Screening Tests	“Screening tests performed with of the synthetic analogs were performed as described earlier ...”	“Screening of the synthetic analogs was performed as described previously...”
6	Screening Tests	“placed in groups of five, at the bottom of 35 ml glass vials.”	“placed in groups of five at the bottom of 35 ml glass vials.”
6	Screening Tests	“Measurements were made using 60µl dose...”	“Measurements were made using 60µl doses...”
7	Data Analysis	“Graphs and statistical analysis using the technique: one way ANOVA...”	“Graphs and statistical analysis using one-way ANOVA...”

7,8	Results and Discussion	Robinson annelation	Robinson annulation
7	Results and Discussion	“Given that the structure of oxalic acid prevails any carbon substitution in the molecule, we chose succinic acid as a proper dicarboxylic substitute...”	“prevails” does not seem to be the correct word here
7	Results and Discussion	Scheme 2	Could be condensed slightly since both 3 and 4 converge to the same products by the same procedure
7	Results and Discussion	“...together with the corresponding corresponded trans-decalol 8...”	“...together with the corresponding trans-decalol 8...”
8,9	Results and Discussion	Scheme 3, 4, and 5	Bond angles/lengths are poorly and unacceptably rendered
8	Results and Discussion	“...construction of the decaline system...”	“...construction of the decalin system...”
8	Results and Discussion	“...in the presence of S-(-)-proline (S-(-)9)...”	“...in the presence of S-(-)-proline and (S-(-)9)...”
8	Results and Discussion	Scheme 4	Yield of final step reads “form 11”, should be “from 11)
8	Results and Discussion	“Should our assumption were correct, the product would be...”	“Should our assumption be correct, the product would be...”
9	Results and Discussion	“Compounds were tested using as basic criterion the ascending stereoselectivity...”	What is meant by “ascending stereoselectivity”?
9	Results and Discussion	“Compound 7, practically a mixture of diastereomers appeared...”	“practically”, which can be interpreted as “almost”, does not make sense; 7 is either a mixture of diastereomers or it is not
11	Conclusion	The main idea behind the above described work was that the structural combination of the two active acaricides would lead to a structure with improved activity against the targeted mites.”	“It was hypothesized that the structural combination of two active acaricides would lead to a structure with improved activity against the targeted mites.”
11	Conclusion		The text is poorly written, and the conclusions/meaning of the data/work are not cogently expressed

Appendix B

Response to the Reviewers' Comments

Manuscript ID: RSOS-200612

Title: “Enantioselective Synthesis of a Costic Acid Analog with Acaricidal Activity against the Bee Parasite *Varroa destructor*”

Journal: *Royal Society of Open Science* (published in collaboration with RSC)

Authors: Georgiladaki, Sofia; Isaakidis, Demosthenes; Spyros, Apostolos; Tsikalas, George; Katerinopoulos, Haralambos, University of Crete
Department of Chemistry

We would like to sincerely thank the reviewers for their suggestions and corrections that helped us to significantly improve the submitted manuscript. In the next sections we address one by one the points raised by the reviewers in the same order they made the comments.

Reviewer: 1

- **Comments and Edits:**

Page #	Section	Text/Figure/Scheme	Correction/Comment	Our Response
1	Summary	“Costic acid is a sesquiterpenecarboxylic acid, present in the plant Dittrichia viscosa .”	Remove comma	Change made according to reviewer's suggestion
1	Summary	“...it is equipotent to costic acid which consists of a transdecalin system with 3 three chiral centers.”	Change “that” to “which” and “3” to “three”	Change made according to reviewer's suggestion

1	Summary	“The basic idea in goal of this project was to design and synthesize a hybrid entity, incorporating aspects of both oxalic acid and costic acid, that would be is that, in spite of the significant differences in structural complexity, the combination of a simple dicarboxylate moiety with the carbon framework of costic acid would lead to an active acaricide. The hypothesis turned out to be correct: the synthetic product was more active than the parent compounds. This approach introduces a useful idea in strategy for the preparation of congeners of bioactive compounds, and in the same time	“The basic goal of this project was to design and synthesize a hybrid entity, incorporating aspects of both oxalic acid and costic acid that would be more active than the parent compounds. This approach introduces a useful strategy for the preparation of congeners of bioactive compounds and proposes a structural framework for a new series of acaricidal agents.”	Sentence modified as suggested by the reviewer.
---	---------	--	--	--

		proposes the a structural framework of a new series of acaricidal agents.”		
1	Introduction	Sentences 1 and 2	Nearly identical to those of the summary	Sentences have been modified
1	Introduction	“varroosis”	Varroosis	Change made according to reviewer’s suggestion
1	Introduction	“an estimated \$8 to \$10 billion worth of crops, only to mention the USDA estimates.”	“an estimated \$8 to \$10 billion worth of crops.” (the last phase does not make sense)	Change made according to reviewer’s suggestion
1	Introduction	“alpha-costic acid, the double bond isomer of costic acid is also acts acting as acaricide against varroa”	“alpha-costic acid, the double bond isomer of costic acid also acts as acaricide against varroa”	Change made according to reviewer’s suggestion

1	Introduction	“analog incorporating structural element of both moieties”	“analog incorporating structural elements of both moieties”	Change made according to reviewer’s suggestion
2	Synthetic Procedures		Faint grey highlighting throughout experimental section?	We were not able to faint the highlighting; we will be in contact with the publisher to make the change in the proofs.
4	Synthetic Procedures	Preparation of compound 10:	2-methylcyclohexanone is defined in parentheses as “mvk”, this is incorrect	Correction made according to reviewer’s suggestion
6	Mites and Bees	“For mite collection two different methods were used: In method A an apparatus introduced by Ariana et al [19] was used.”	“For mite collection two different methods were used: in method A an apparatus introduced by Ariana et al [19] was used.”	Change made according to reviewer’s suggestion
6	Mites and Bees	“The inner cylinder was taken out of the apparatus to return the bees to their mother colony; a few minutes after they had recovered from the effect of the anesthesia.”	“The inner cylinder was taken out of the apparatus to return the bees to their mother colony a few minutes after they had recovered from the effect of the anesthesia.”	“The inner cylinder was taken out of the apparatus to return the bees to their mother colony. A few minutes later they all recovered from the effect of the anesthesia”.
6	Mites and Bees		It is not clear how the two methods of mite collection are different as the text states that soft brushes and stereoscopes are used to separate the mites and bees in both cases	The correct statement is: The bottom lid of the outer cylinder was taken off and the mites were collected and placed into the test vials.
6	Screening Tests	“Screening tests performed with of the synthetic analogs were performed as described earlier ...”	“Screening of the synthetic analogs was performed as described previously...”	Change made according to reviewer’s suggestion
6	Screening Tests	“placed in groups of five, at the bottom of 35 ml glass vials.”	“placed in groups of five at the bottom of 35 ml glass vials.”	Change made according to reviewer’s suggestion
6	Screening Tests	“Measurements were made using 60µl dose...”	“Measurements were made using 60µl doses...”	Change made according to reviewer’s suggestion
7	Data Analysis	“Graphs and statistical analysis using the technique: one way ANOVA...”	“Graphs and statistical analysis using one-way ANOVA...”	Change made according to reviewer’s suggestion

7,8	Results and Discussion	Robinson annelation	Robinson annulation	“annelation” is also used, however we changed it to annulation, since the latter term is more often used
7	Results and Discussion	“Given that the structure of oxalic acid prevails any carbon substitution in the molecule, we chose succinic acid as a proper dicarboxylic substitute...”	“prevails” does not seem to be the correct word here	“prevents” is the correct word
7	Results and Discussion	Scheme 2	Could be condensed slightly since both 3 and 4 converge to the same products by the same procedure	The Scheme was modified regarding compounds 3a, 3b, 4a & 4b
7	Results and Discussion	“...together with the corresponding corresponded trans-decalol 8...”	“...together with the corresponding trans-decalol 8...”	Change made according to reviewer’s suggestion
8,9	Results and Discussion	Scheme 3, 4, and 5	Bond angles/lengths are poorly and unacceptably rendered	Schemes redrawn/corrected, please see comment on reviewer 2
8	Results and Discussion	“...construction of the decaline system...”	“...construction of the decalin system...”	Change made according to reviewer’s suggestion
8	Results and Discussion	“...in the presence of S-(-)proline (S-(-)9)...”	“...in the presence of S-(-)-proline and (S-(-)9)...”	“...in the presence of S-(-)-proline (abbreviated as S-(-)9)...”
8	Results and Discussion	Scheme 4	Yield of final step reads “form 11”, should be “from 11)	Change made according to reviewer’s suggestion
8	Results and Discussion	“Should our assumption were correct, the product would be...”	“Should our assumption be correct, the product would be...”	Change made according to reviewer’s suggestion
9	Results and Discussion	“Compounds were tested using as basic criterion the ascending stereoselectivity...”	What is meant by “ascending stereoselectivity”?	Sentence was deleted...
9	Results and Discussion	“Compound 7, practically a mixture of diastereomers appeared...”	“practically”, which can be interpreted as “almost”, does not make sense; 7 is either a mixture of diastereomers or it is not	“Compound 7, which is a mixture of diastereomers appeared...”

11	Conclusion	The main idea behind the above described work was that the structural combination of the two active acaricides would lead to a structure with improved activity against the targeted mites.”	“It was hypothesized that the structural combination of two active acaricides would lead to a structure with improved activity against the targeted mites.”	Change made according to reviewer’s suggestion
11	Conclusion		The text is poorly written, and the conclusions/meaning of the data/work are not cogently expressed	The Conclusions section was re-written to make a clear statement of the initial idea and the conclusions made, based on the evaluation of the experimental data

Critical Evaluation:

Comment No	Comment	Our response
1	It seem plausible that in addition to the alkene isomers shown (3a, 3b, 4a, 4b), that the conjugated acrylate and fumarate isomers are plausible. These would be access by a similar isomerization mech via enolate intermediates. Qualitatively, it seems that the acrylate/fumarate would also be thermodynamically preferred	We thank the reviewer for pointing out the possibility of double bond isomers. We have excluded the possibility of the acrylate analog since, in this case the double bond is tetrasubstituted; the ¹HNMR spectrum though, indicates the presence of a vinylic proton. In 4a,b for instance, this proton appears at 5.51-5.46 (m, 1H), and the corresponding carbon appears at 124.5/124.0 (two diastereoisomers) in agreement with theoretical predictions for the structures shown. If the isomer were the fumarate, the corresponding values would have been (approximately) 6.2 ppm and 129 for the proton and carbon resonances, respectively.
2	The a-stereocenter is not defined in any of the structures (4, 5, 6, 7). Further, in cases where a diastereomeric ratios can be determined, it should be. Further, inspection of the NMR such as certain versions of 6 and 7 appear to be single diastereomers. Thus, the stereochemistry at the a-position needs to be defined.	Unfortunately, the synthetic schemes followed, and the nature of the final products do not allow the determination of the absolute configuration of the a-stereocenter: Stobe reaction conditions are not enantioselective. Moreover the alpha proton in (2R,4aR,8aR)-6 and (2R,4aR,8aR)-7, is very acidic and this precludes any attempts to modify the structures in strong basic or acidic conditions that would create enolate or enol intermediates. Mosher ester analysis would require the reduction of the carboxylates to the corresponding alcohols under conditions that would epimerize the stereocenter. We believe that the active configuration of the alpha-center would be revealed through an enantioselective synthesis of a tricyclic analog that would secure the formation of one enantiomer. A similar comment has been added to the end of the discussion section. Regarding diastereomeric ratios, both de and ee were determined for (2R)-2 and (2R,4aR,8aR)-6 using chiral GC or chiral HPLC analyses (see coment of reviewer 2)

	While not ultimately prohibitive to the reader's understanding of the text, the numerous grammatical hamper the reading progression... please closely edit this manuscript an additional time before resubmission.	The entire document has been edited and modified grammatically and syntactically. Changes in the text are in compliance with the reviewer's suggestions
4	The work demonstrates how classic, well-established chemistry (e.g. Claisen condensations, dissolving metal reduction, Robinson annulation) can be employed to access novel, valuable frameworks for preparing valuable lead molecules.	We thank the reviewer for the comment. Our intention was to demonstrate that, if a synthetic compound were chosen over an active natural product, its preparation should be simple using readily available starting materials

Reviewer: 2

Comment No	Comment	Our response
1	The Authors describe this chemistry as an enantioselective synthesis, but there does not seem to be an ee for all chiral compounds - this needs to be rectified.	Both de and ee were determined for (2R)-2 and (2R,4aR,8aR)-6 using chiral GC or chiral HPLC analyses. The choice of analytical method was based on the resolution of the pure enantiomers and the diastereomeric mixtures. Details are given in the supplementary information file.
2	Yields should be included in schemes	Yield are included in all steps.
3	HRMS data should be given over low resolution mass spec	Exact mass analyses was performed for all compounds which are not mixtures or non-isolated intermediates (please see Electronic Supplementary Information).
4	LCMS traces should be given as a further indication of purity (or elemental analysis)	Adopting the RSC Associated Editor's advice on this suggestion, we provided copies of ¹ HNMR (as well as other 1D or 2D NMR) spectra as a valid alternative to showing compound purity. Furthermore we performed exact mass analyses for all compounds which are not mixtures or non-isolated intermediates (please see Electronic Supplementary Information Material).
5	The structure of compound 12 does not seem to be correct	It was correct... It was drawn in a way to indicate that a six membered ring will be formed at the next step. However, there was not enough space for the methyl to be drawn. We therefore, depicted structures 11 and 12 with extended side chain.
6	No need to include the characterisation data for the compounds in the manuscript - these belong in the SI	NMR and MS data were transferred to the SI.